# A rapid realist review of universal interventions to promote inclusivity and acceptance of diverse sexual and gender identities in schools

Merle Schlief [1,2], Theodora Stefanidou [1,2], Talen Wright[1], Grace Levy[1], Alexandra Pitman[1] & Gemma Lewis [1]✉

Universal interventions to promote inclusivity and acceptance of diverse sexual and gender identities in schools could help to prevent mental health problems in this population. We reviewed evidence and developed programme theories to explain which universal interventions work, for whom, in which contexts and why. We conducted a rapid realist review and extracted data in context–mechanism–outcome configurations, to develop and refine programme theories. We included 53 sources and identified five intervention themes: student pride clubs, inclusive antibullying and harassment policies, inclusive curricula, workshops and staff training. Here, we show that these interventions could work by reducing discrimination, bullying and marginalization. Interventions appear to work best when school staff are trained and the school climate is supportive and may be less effective for boys, gender minority students and bisexual students. Our findings provide guiding principles for schools to develop interventions and should encourage primary research to confirm, refute or refine our programme theories.

Depression and anxiety are common mental health problems, which often begin during adolescence[1,2]. Self-harm is frequently comorbid with adolescent depression and anxiety and these mental health problems are leading risk factors for suicide[3,4]. There is evidence that rates of depression, anxiety, self-harm and suicide are rising among young people[1,5,6]. Public health interventions to prevent these mental health problems would reduce their rising incidence and alleviate the burden on clinical services.

Sexual and gender minorities (SGM, people who are not heterosexual or cisgender) are often exposed to stigma, prejudice, discrimination and abuse within societies that promote being heterosexual and cisgender as normal[7–9]. Minority stress theory proposes that stigma, prejudice and discrimination create a hostile and stressful environment that causes mental health problems among sexual and gender minorities. Sexual minority young people (including but not limited to those who are lesbian, gay, bisexual or queer) are twice as likely to experience depression, anxiety, self-harm and suicidality than their heterosexual peers[10–12]. There are few high-quality population-based studies of mental health among gender minority (including but not limited to those who are transgender, non-binary and gender diverse) compared with cisgender young people[13,14]. However, there is evidence that gender minority young people are at increased risk of depression, anxiety, self-harm and suicidality[15–19].

Universal interventions aim to reduce exposure to modifiable causal risk factors and have succeeded at preventing heart disease and certain cancers[20,21]. Universal interventions could transform the

[1]Division of Psychiatry, Faculty of Brain Sciences, University College London, London, UK. [2]These authors contributed equally: Merle Schlief, Theodora Stefanidou. ✉e-mail: gemma.lewis@ucl.ac.uk

prevention of mental health problems[20] but their development continues to lag behind those for physical health. Schools are a potential setting for preventative interventions that would reach most young people. There is evidence that, in schools, SGM young people experience higher levels of bullying, discrimination, exclusion and marginalization than their heterosexual or cisgender peers[10,22–25]. Universal interventions which promote inclusivity and acceptance of diverse sexual and gender identities in schools could prevent or reduce mental health problems among SGM young people.

To our knowledge, no study has synthesized evidence on universal school-based interventions to promote inclusivity and acceptance of diverse sexual and gender identities. In addition to identifying interventions, it is important to investigate which work, in what context, for whom and how to inform effective implementation. The effectiveness of interventions might depend on the contexts in which they are implemented. For example, some interventions might work better in schools with already high acceptance and inclusivity while being potentially harmful in schools with lower acceptance and inclusivity. Similarly, interventions might increase acceptance and inclusivity towards sexual but not gender minorities.

Realist reviews use context–mechanism–outcome (CMO) configurations to generate programme theories, which suggest that certain interventions are more or less likely to work, for certain people, in certain situations[26]. The aim is to develop, refine and test theories about how interventions interact with contexts (C, people and environments), by triggering mechanisms (M, internal psychosocial reactions and reasonings) to generate outcomes (O)[26,27]. Realist approaches to evidence synthesis can be used to complement systematic reviews and meta-analyses, by providing evidence beyond effectiveness[28,29].

Rapid realist reviews enable evidence to be produced in a timely and resource-sensitive manner for policy decision-making[26]. Compared to traditional realist reviews, rapid realist reviews do not aim to conduct comprehensive literature searches of peer-reviewed literature and other sources. Instead, they draw more heavily on input from reference groups and include experts in research and practice to develop programme theories, accelerate the reviewing process and advise on the dissemination and use of findings[26]. The rapid realist methodology has previously been used to provide timely evidence in a range of areas, including school-based interventions and healthcare[27,28,30]. We conducted a rapid realist review to investigate the following questions:

(1) What universal school-based interventions exist to promote inclusivity and acceptance of diverse sexual and gender identities and how and where were they implemented?
(2) In which contexts, and for whom, do these interventions work (or not work) and why?

## Results

We identified 5,155 records from database searches and 16 through other sources including the call for evidence, websites and reference checking (Fig. 1). We screened 407 full texts and included 53 eligible sources (Fig. 1): 52 peer reviewed and one other source[31]. All included sources were relevant to the development of the programme theories. The rigour of sources was mixed: the methods of 22 of 53 sources were deemed trustworthy and credible, that is the data collected allowed the study to address the research question and the authors' interpretation of the results was substantiated by their data (Supplementary Table 3). Twelve sources did not fulfil either of these criteria and the remaining 19 sources were somewhat trustworthy and credible or the rigour of sources was considered unclear. Detailed information on the search strategy, inclusion and exclusion criteria and quality assessment can be found in the Methods.

Of the peer-reviewed papers, 6 used mixed methods, 14 were qualitative and 25 were quantitative. Of the quantitative studies, 17 were cross-sectional, 5 used prepost comparison designs,

2 were cohort studies and 1 was a randomized controlled trial (RCT). We also included 6 reviews and, following realist guidelines, 1 non-peer-reviewed source.

Sources were published between 1995 and 2021 and 65% were conducted in North America (Table 1). Thirteen included data on mental health outcomes (Supplementary Table 5). Study characteristics are presented in Supplementary Table 4.

We classified interventions into five themes (Supplementary Table 6): gay–straight alliances (GSAs) and similar student clubs (for example, pride clubs); inclusive antibullying and harassment policies; inclusive curricula; workshops including media-based interventions and LGBTQ+ ally training.

Our initial programme theory proposed that strategies to promote inclusivity and acceptance for SGM young people in schools would reduce their risk of depression, anxiety, self-harm and suicidality. This was generally supported within each of our themes. We present refined programme theories for each theme separately (Table 2 and Figs. 2–5). Some themes had multiple programme theories, to represent distinct outcomes or mechanisms. Each theme includes additional information around contexts, mechanisms and potential harms. Where a CMO was raised by or strongly supported by the Young Person's Advisory Group (YPAG) or Stakeholder's Advisory Group (SAG), we reference 'YPAG' or 'SAG.' Individual CMOs and references for each section are provided in Supplementary Tables 7–11. Detailed information on the development of the programme theories can be found in the Methods.

### Gay–straight alliances and similar student clubs
**Programme theories.** When SGM students attend schools with GSAs or similar clubs (C), they may experience reductions in bullying and discrimination (O) (Fig. 2). This could be because these clubs reduce homophobia, biphobia and transphobia, improve relationships between students, empower SGMs to speak out, validate being lesbian, gay, bisexual, trans and queer (LGBTQ+) and improve school climate (M) (SAG)[32–38].

When SGM students attend schools with GSAs or similar clubs (C), they report reductions in suicidal thoughts and attempts, improvements in academic performance, increased school attendance, reductions in isolation and increased feelings of safety (O). This could be because of reductions in bullying and increases in social support and connectedness, due to safe spaces where students make friends, validate their thoughts and feelings, do not feel judged and build positive relationships with school staff (M) (SAG and YPAG)[32–35,37–40].

**Additional information on mechanisms.** When teachers who identify as sexual or gender minorities also attend GSAs and similar clubs, it may enhance their positive impact because students are exposed to role models who they can turn to for support (SAG and YPAG)[39]. Staff can communicate their support by attending GSAs or wearing rainbow lanyards (YPAG). The longer-established the GSA or similar club, the more likely it is to be effective[35,36]. It is also important that GSAs and similar clubs are taken as seriously as other clubs (YPAG).

**Key contexts and groups.** Young people who are still coming to terms with their sexual orientation or gender may not attend GSAs or similar clubs. However, the presence of a GSA or similar club could be more important than participating in it, perhaps because the activities benefit the whole school[32]. Setting up a successful GSA might depend on school climate including openness amongst students and staff, a whole-school 'inclusivity' approach as well as tailoring for the school's demographics and ethos (SAG). Resistance and ignorance from parents, conservatism in families, lack of confidence or skills in teachers as well as single-sex boys' schools can be barriers to successfully implementing GSAs (SAG). One study found that although GSAs reduced bullying and improved feelings of safety, there was no reduction in depressive symptoms[32]. Reasons for this finding were unclear.

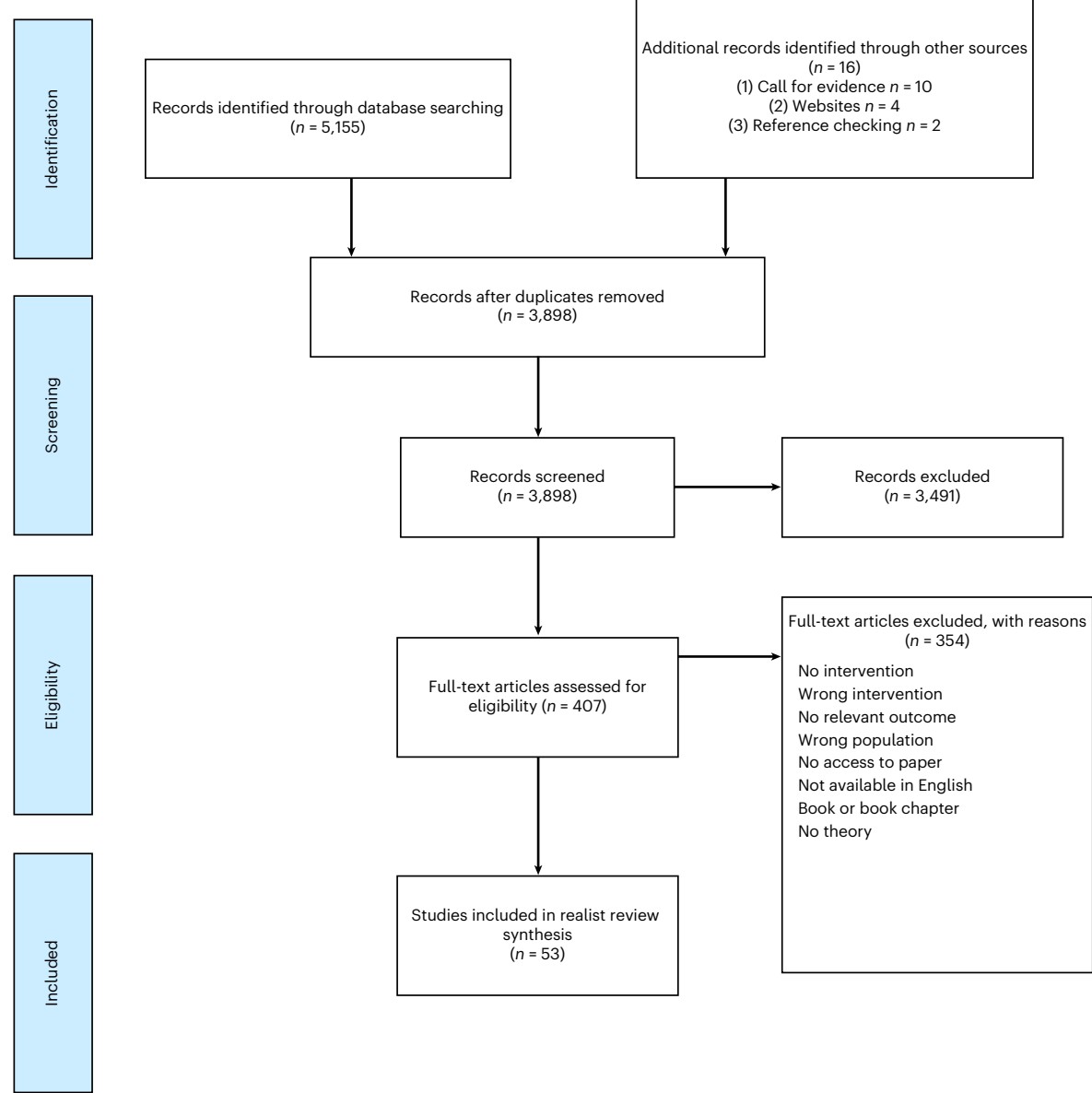

**Fig. 1 | PRISMA 2009 flow diagram.** Flow diagram generated according to PRISMA[77].

**Potential for harm.** If the wider school environment is not supportive, GSAs or similar clubs could increase bullying because the visibility of SGM students is increased[39,41]. SGM students might be reluctant to attend GSAs or similar clubs if they fear being stigmatized and bullied for attending them (SAG and YPAG). This might particularly be the case in rural settings[41]. Members of GSAs or similar clubs, might also become isolated from the wider school community (YPAG). The wider school context could be addressed and the climate of a school assessed first, to determine what type of intervention might be most effective (YPAG and SAG). Our SAG also suggested that, beyond the wider environment, if a GSA is not run well then it might not be a safe space for all members (SAG).

**Inclusive antibullying and harassment policies**
**Programme theories.** When SGM students attend schools with inclusive antibullying and harassment policies, and staff are aware of these policies and implement them (C), students feel safer, have higher self-esteem and are less likely to experience self-harm, suicidal ideation, suicide attempts and absenteeism (O) (Fig. 3). This could

be because of reduced bullying and homophobic aggression[42,43] and a more supportive school culture, with staff and students likely to intervene (M)[34,35,38,42–47].

When school staff implement inclusive policies in rural or politically conservative communities, with religious groups that oppose equal rights (C), they may face barriers such as unsupportive school leadership, patriarchal values and hetero- and cis-normativity (O), due to lack of systemic changes to attitudes (M)[48–50].

When inclusive antibullying policies address homophobic language within broader conversations about social status, popularity and masculinity (C), this is more likely to reduce homophobic slurs (O). This could be because heterosexual students often do not see themselves as homophobic but understand ideas about popularity and masculinity (M)[51].

**Additional information on mechanisms and strategies.** It is important that policies are supported by school leaders and the implementation of policies is monitored. If schools have processes in place to record incidents of homophobic, biphobic and transphobic

**Table 1 | Summary of studies (*n*=53)**

| Study setting | | Study design | |
|---|---|---|---|
| Country | Number of studies | Design | Number of studies |
| United States | 27 | Quantitative cross-sectional studies | 17 |
| Canada | 7 | Quantitative non-randomized trials/cohort studies | 2 |
| Australia | 5 | Quantitative observational studies with prepost comparator only | 5 |
| England | 2 | RCT | 1 |
| South Africa | 2 | Qualitative | 14 |
| New Zealand | 2 | Mixed methods | 6 |
| the Netherlands | 2 | Systematic review | 4 |
| Italy | 1 | Other review | 2 |
| Taiwan | 1 | Other | 2 |
| Philippines | 1 | | |
| Norway | 1 | | |
| Israel | 1 | | |
| Republic of Ireland | 1 | | |

bullying, students and teachers might be more likely to report these incidents (SAG).

**Key contexts and groups.** It is possible that lesbian and gay but not bisexual or gender minority (bi SGM) students are at reduced risk of bullying and suicide attempts in schools with inclusive antibullying policies compared to those without[44]. This might be because risk factors are different among bi SGM, compared with gay and lesbian young people[43,44]. The positive effects of inclusive school policies might be less persistent among boys/young men than girls/young women[52]. It seems necessary that the school policy is an LGBTQ+ inclusive one, not just a general one, as these do not reduce bullying among SGM students[44].

**Potential for harms.** Gender equity government legislation aims to address gender inequity in schools. When gender equity policies are implemented in schools that are hostile to sexual and gender minorities, these students might experience increases in bullying or isolation[49]. Students might gain a false sense of safety and face backlash when being 'out' about their sexuality or gender[53]. Our YPAG proposed conflict resolution talks to address bullying instead of punishments such as detention, which do not educate the perpetrators. They also suggested that safeguarding issues should be evaluated to respect the privacy of SGM students (YPAG) when reporting incidents. Information about students' sexual or gender identity should not be revealed to parents/carers (YPAG)[53].

**Inclusive curricula**
**Programme theories.** When schools have inclusive curricula, with positive representation of SGMs (C), SGM students are less likely to be bullied and other students are more likely to intervene (O1) (Fig. 4). This can improve connectedness (O2) as well as self-esteem and well-being and reduce suicidal ideation among SGM students (O3). This could be because inclusive curricula increase awareness, understanding and acceptance (M1), validate sexual and gender minorities (M2), oppose compulsory heterosexuality (M3) and improve school climate (M4)[31,33,39,54–66].

**Additional information on mechanisms and strategies.** Inclusive curricula seem most effective when they: avoid 'deficit and at-risk narratives', make the contributions and achievements of LGBTQ+ role models visible, use workbooks and literature that include LGBTQ+ issues, facilitate indepth reflection on LGBTQ+ topics beyond learning facts, have sticker systems to highlight books with LGBTQ+ themes and/or characters, include LGBTQ+ topics in sexual health education and are implemented from an early age onwards (SAG and YPAG)[39,54–61]. Inclusive curricula should be codesigned and codelivered by teachers and LGBTQ+ students (SAG). Our YPAG stated that students should be better educated on the history of LGBTQ+ people, for example the lesbian community providing activism and support during the HIV/AIDS crisis in the 1980s and 1990s. Our SAG suggested that external speakers such as mental health professional and human rights activists can provide additional insights into the challenges LGBTQ+ people experience.

**Key contexts and groups.** Inclusive curricula seem particularly effective for students who are severely victimized on the basis of gender expression or in schools with hostile climates[33,60]. Not all studies found reductions in bullying and victimization after implementing inclusive curricula[31,59]. While it is unclear what the mechanisms of these differential effects are, it might be due to school climates and ingredients of curricula.

**Potential for harms.** When inclusive curricula face a backlash from the wider community, they might lead to increased bullying of SGM students[50]. Our SAG suggested that schools might face pushback from parents who are opposed to inclusive curricula. If teachers are not well-informed on LGBTQ+ issues, they might not address topics sensitively and use incorrect language and/or pronouns (SAG and YPAG). They might fear unintentionally causing offence (SAG).

**Workshops including media interventions**
**Programme theories.** When students attend workshops on sexual and gender diversity, led by sexual and gender minorities or assemblies or media interventions led by SGM students (C), this increases inclusivity and acceptance, decreases bullying and increases the likelihood of students intervening (O) (Fig. 5). This could be because workshops increase students' understanding and acceptance, promote empathy and raise awareness of the harmful effects of discrimination (M) (YPAG)[62–67].

**Additional information on mechanisms and strategies.** Peer educators with lived experience seem to play an important role in increasing inclusivity and acceptance and reducing bullying[62,64,65]. Interventions might be particularly effective if they provide information on how to be an ally and how to behave when witnessing bullying and harassment[66]. Young students might especially benefit from workshops and media interventions, as this can foster acceptance and inclusion from a young age (SAG). However, one study in the Netherlands found mixed effects of a peer intervention on attitudes and bullying among male students. This might be due to the content of the intervention, the school context and/or the age of students[68]. Workshops should not be tokenistic (for example, occurring during pride month but not thereafter) and should be part of a meaningful, long-term commitment including different school interventions (SAG and YPAG).

**Potential for harms.** In a study conducted in the Netherlands, there was some evidence that positive attitudes towards SGM students and willingness to intervene declined after a peer-led intervention, particularly among male students. This could have been due to the content and nature of the intervention as well as the school context[68].

**Table 2 | Summary of CMO configurations comprising the programme theory for each intervention theme**

| Type of intervention | Context (when the intervention works best) | Mechanism (why the intervention works) | Outcome | For whom |
|---|---|---|---|---|
| GSAs or similar student clubs (for example, pride clubs) | (1) Longer-established clubs (2) Clubs integrated in wider school strategy (3) Schools with positive climate (4) LGBTQ+ teachers attending the clubs and wearing rainbow lanyards | Reduced homophobia, improved relationships between students, empower SGM students, normalization of being LGBTQ+ → improved school climate | Reduction in self-reported bullying and discrimination | SGM |
| | | Reduced bullying and safe space for self-expression and social activities | Reduced likelihood of suicidal thoughts and attempts; reduced isolation and increased feelings of safety | SGM students |
| Inclusive antibullying and harassment policies | (1) Longer-established policies (2) Policies being specific to LGBTQ+ issues (3) Supportive school leadership (4) Staff being aware and implementing policies (5) Education and support to bullies (6) Combination of multiple policies in least safe schools | Reduced homophobia → reduced bullying and stressors → improved school climate | Increased feelings of safety and higher self-esteem; reduced likelihood of self-harm, suicidal thoughts and attempts | SGM students; differential effects for lesbian, gay and bi SGM students |
| Workshops including media interventions | (1) Workshops held by LGBTQ+ peer educators (2) Media interventions led by LGBGTQ+ students (3) Included in a wider long-term commitment to inclusivity and acceptance by the school | Increased empathy and understanding towards LGBTQ+ students; awareness of discrimination | Increased inclusivity and acceptance; decreased homophobic and transphobic bullying | SGM students |
| LGBTQ+ ally and staff training | (1) Training on how to discuss homophobic language use and bullying (2) Sufficient training and resources (3) Training codesigned and codelivered by LGBTQ+ staff and students | Staff more equipped to implement interventions, provide support and be inclusive towards LGBTQ+ students → increased acceptance, support, treatment, connection and safe learning environments | Less victimization; greater self-esteem, well-being and mental health | SGM students |
| | | Increased likelihood of discussing, responding to and intervening with homophobic language use and bullying | Increased likelihood of feeling safe and less victimized | SGM students |
| Inclusive curricula | (1) Positive LGBTQ+ representation/role models (2) Avoiding 'deficit and at-risk' narratives (3) Education on LGBTQ+ issues (4) Implementation at an early age | Increased understanding of experiences of LGBTQ+ people, including bullying → acceptance and normalization of being LGBTQ+ and improved school climate | Decreased victimization and bullying and increased intervention with bullying | SGM students, especially severely victimized students |

Key: GSA, gay–straight alliances; LGBTQ+, lesbian, gay, bisexual, trans and queer; SGM, sexual and gender minority; →, is hypothesized to lead to.

**LGBTQ+ ally and staff training**

**Programme theories.** When teachers and school staff are well-informed about sexuality and gender issues (C), SGM students experience less victimization, greater self-esteem, improved mental health, fewer days of school absence and higher attainment (O) (Fig. 6). This could be because staff are better equipped to create safe spaces, support GSAs and inclusive curricula and refer students to community and counselling support (M1). Students are also likely to build connections and feel accepted within a safe and progressive environment where gender binary norms are challenged and staff use correct pronouns (M2)[33,56,69–71].

When teachers receive training in how to be an ally, which provides them with information about language and behaviour (C), SGM students feel safer and less victimized (O). This could be because teachers and students are more likely to discuss, respond to and intervene against such behaviour (M)[41,69–74].

**Additional information on mechanisms and strategies.** One of the barriers to school staff supporting SGM students is insufficient training and resources, including lack of knowledge about pronouns (SAG and YPAG)[70]. Teachers might be more likely to discuss homophobic language in class but not more likely to intervene after a training course,

if not sufficiently prepared to do so[72]. Training on LGBTQ+ topics might be particularly effective if codesigned and codelivered by teachers and LGBTQ+ students (SAG and YPAG).

## Discussion

We identified five types of universal intervention designed to promote inclusivity and acceptance of diverse sexual and gender identities in secondary schools. Interventions included GSAs or similar student clubs (for example, pride clubs), LGBTQ+ inclusive antibullying and harassment policies, LGBTQ+ inclusive curricula, workshops including media-based interventions and LGBTQ+ ally and staff training. We produced a conceptual framework (programme theory) to explain how these interventions might work, for whom, in which contexts and why. Our findings supported our initial programme theory, which proposed that improving inclusivity and acceptance for SGM young people in schools would reduce their risks of depression, anxiety, self-harm and suicidality. Our findings further elucidated the role of specific contexts and mechanisms underlying the potential impact of universal school-based interventions.

Several studies found evidence that GSAs or similar student clubs were associated with reductions in bullying and improvements in mental health among students. Our programme theory suggested

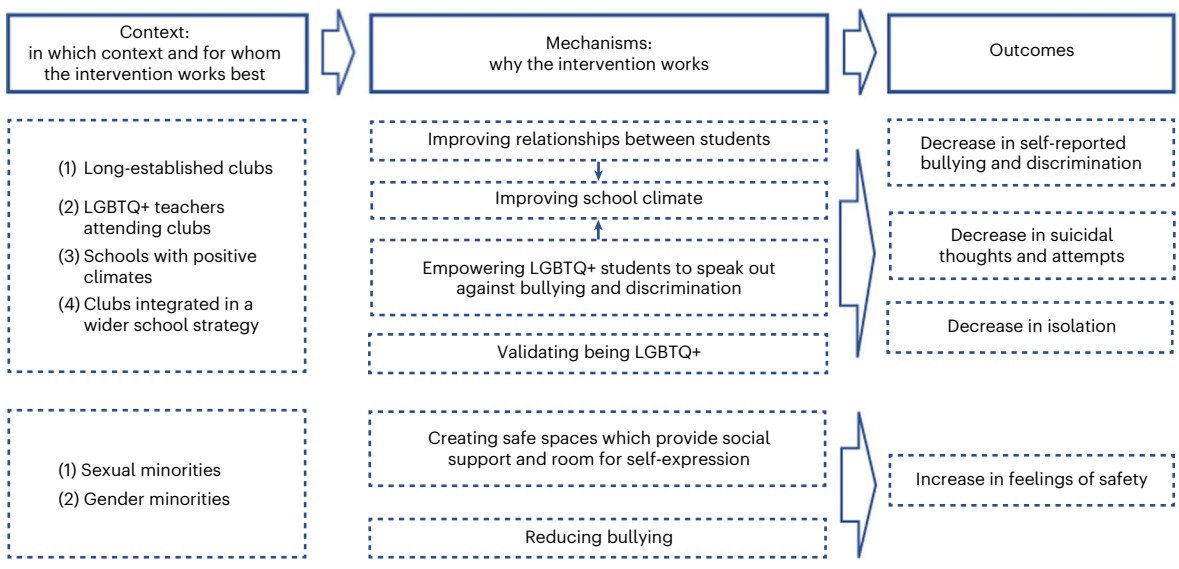

**Fig. 2 | Programme theory for GSAs and similar student clubs (for example, pride clubs).** Figure shows CMO configurations.

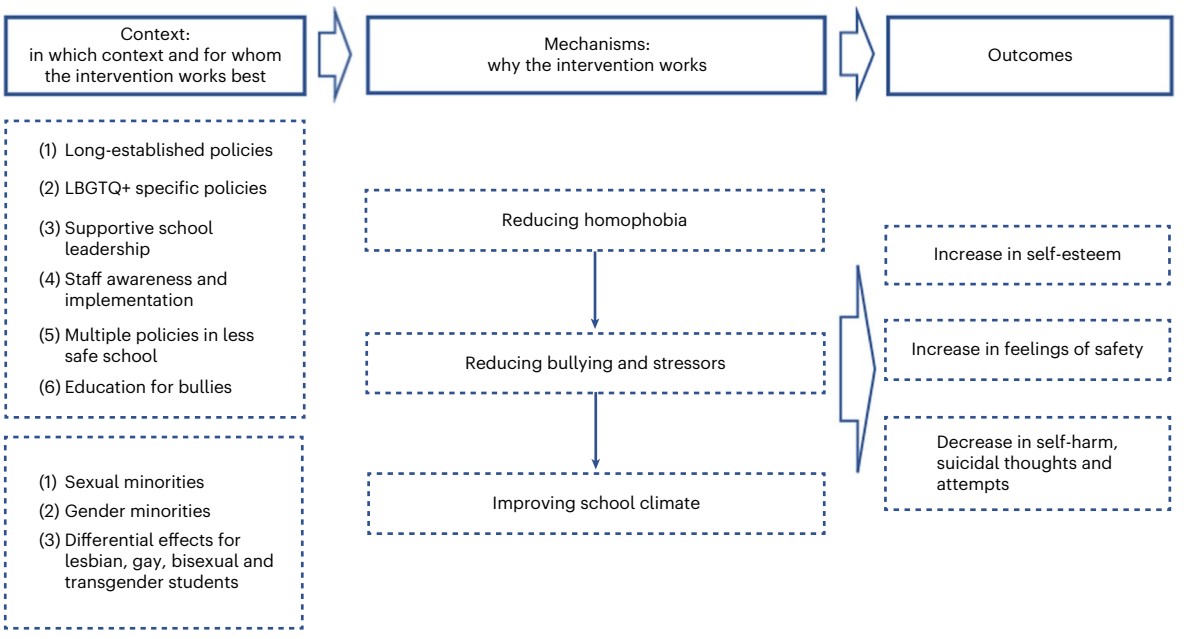

**Fig. 3 | Programme theory for inclusive antibullying and harassment policies.** Figure shows CMO configurations.

that GSAs or similar student clubs seemed to perform better when they were longer-established and attended by teaching staff who were sexual or gender minority role models. The potential benefits of GSAs or similar student clubs might depend upon the pre-existing school climate. These clubs are likely to make SGM students more visible, which could increase their exposure to bullying and discrimination. It is therefore possible that GSAs and similar clubs tend to be implemented, and continued longer-term, in schools with more positive climates. The school climate emerged as particularly important in our review. School climate is shaped by norms, beliefs, relationships (within the school and with the community), teaching and learning practices and the organizational and physical features of the school[74]. As school-level approaches, inclusive curricula and antibullying and harassment policies might be more effective at changing the school climate than GSAs or similar student clubs. However, these three approaches to intervention seem complementary.

Inclusive antibullying and harassment policies may be less effective for bi SGM than for lesbian or gay students. These policies may need to be adapted so they are effective for these young people. The existence of inclusive antibullying and harassment policies may not be sufficient to reduce discrimination and harassment towards SGM students. Implementation seems to depend upon the awareness of teaching staff and the active support of school leaders and the wider community. Inclusive antibullying and harassment policies could work best when there is education and support for bullies (for example, restorative justice) and a combination of multiple policies, particularly in the least safe schools.

Inclusive curricula seem to work best when there is implementation at an early age and positive representation of the achievements and contributions of SGM role models. Inclusive curricula could avoid focusing on 'deficit and at-risk' narratives and validate sexual and gender minorities as being equal to heterosexual and cisgender people.

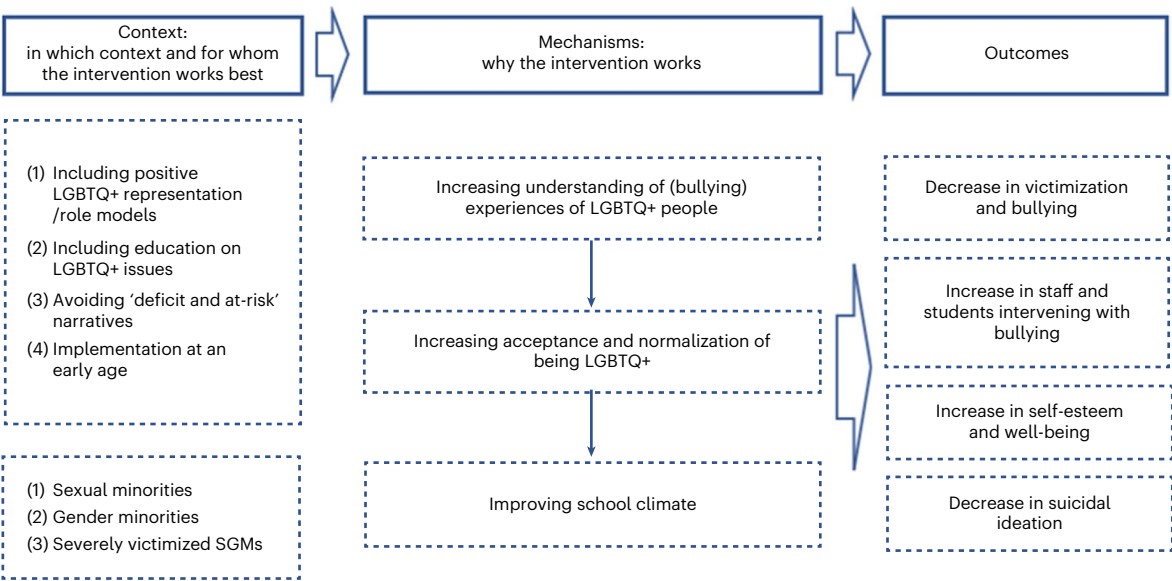

**Fig. 4 | Programme theory for inclusive curricula.** Figure shows CMO configurations.

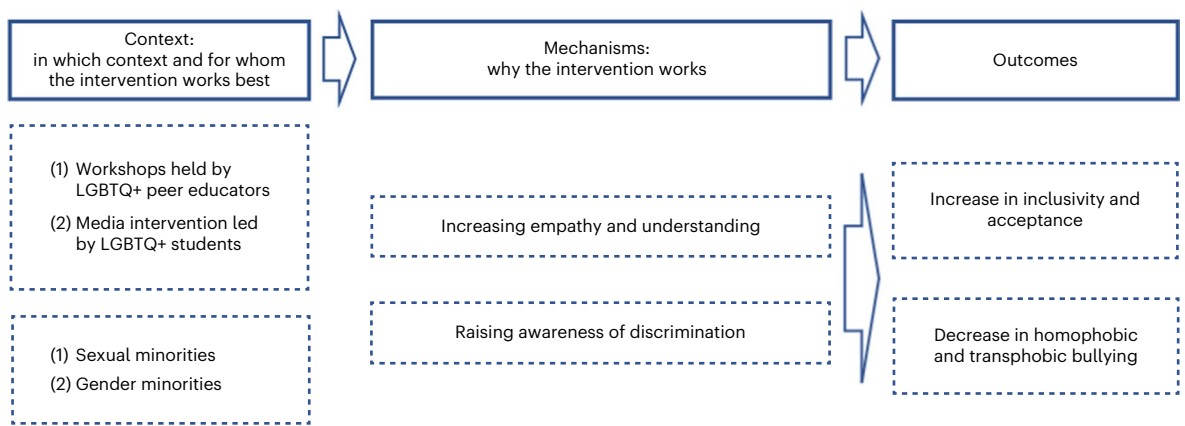

**Fig. 5 | Programme theory for workshops including media interventions.** Figure shows CMO configurations.

Inclusive curricula could benefit all SGM students, especially those who have experienced severe victimization.

Representation of SGM role models emerged as an important theme in our review. For example, workshops and media interventions might be more effective when they are led by people who are SGMs. This could increase empathy, awareness and understanding and lead to increased inclusivity and acceptance.

It might be harder to reduce homophobia, biphobia and transphobia among boys and young men compared with girls and young women. This is perhaps consistent with evidence that women are less likely to hold negative attitudes towards sexual minorities than are men[75]. Universal interventions in schools could be adapted for boys to focus less on the terms homophobia, biphobia and transphobia and, instead, challenge issues of masculinity and popularity.

Our literature search was systematic but, consistent with recommendations for rapid realist reviews, we did not aim to capture all studies exhaustively[26]. We assessed the rigour and relevance of each source to our programme theories, which were the main outputs of our investigation. Our programme theories were informed, refined and endorsed by experts by lived experience, including young people, teachers, policy representatives and school governors. This should improve the validity and generalizability of our theories and the relevance and feasibility of our recommendations for policy and practice.

Although our initial programme theory was generally supported, few studies reported data on depression and anxiety. Several studies reported data on self-harm and suicidality. Interventions that reduce the risk of self-harm and suicidality could also be associated with reductions in depression and anxiety, but more research on this is needed.

Most studies were conducted in North America or Australia. Findings from these countries are unlikely to generalize to other settings, particularly low- and middle-income countries. Few studies were large enough to meaningfully distinguish between SGM groups. We also found little evidence on whether the effectiveness of interventions varied according to factors such as age, ethnicity or symptom severity. While all included sources were relevant to the development of our programme theories, only 22 of 53 sources described methods that were considered trustworthy and credible. We found only one RCT[59].

The school climate emerged as particularly important for the implementation and potential impact of universal school-based interventions. Implementing multiple universal approaches could maximize the possibility of changing the school climate and improving outcomes for students. The order in which interventions are implemented could also be considered. Inclusive curricula and antibullying and harassment policies could be implemented before GSAs or similar clubs. This would demonstrate that the school promotes inclusive and accepting attitudes towards SGMs and does not tolerate bullying based on these

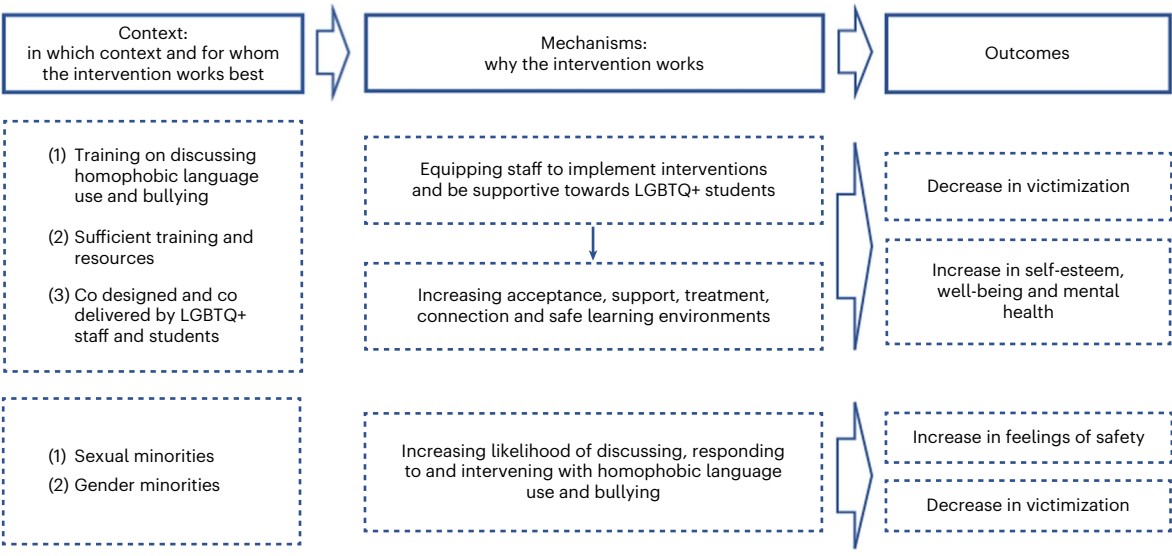

**Fig. 6 | Programme theory for LGBTQ+ ally and staff training.** Figure shows CMO configurations.

characteristics. The clubs would therefore be supported by a wider movement within the school, at the policy level, with the support of senior leadership. It seems important that all school staff are made aware of these school policies and implement them, and that there are processes for reporting homophobic/biphobic/transphobic incidents.

Universal interventions including inclusive curricula, staff training and workshops should be codeveloped and delivered by SGM students, staff, experts by lived experience and peer educators. Schools should promote the positive representation of diverse sexual and gender identities at an early age through inclusive curricula and SGM teachers and school staff attending GSAs and similar clubs.

The implementation and effectiveness of all interventions is likely to depend on how well-trained teaching and school staff are. Sufficient teacher training and resources could be provided so that teachers and school staff are educated to be aware of, and feel confident at challenging, slurs and bullying. School staff might then be better equipped to implement interventions, provide support and be inclusive towards SGM students. This could lead to increased acceptance, support and safer learning environments. In turn, this could reduce bullying and improve mental health for SGM students.

Further research should include RCTs that examine the effectiveness of interventions. Studies should also systematically evaluate the implementation and impact of specific aspects of interventions to elucidate the contexts and mechanisms of successful interventions. Further evidence is needed on the potential impact of universal school-based interventions on mental health outcomes among gender minority young people and different sexual minority students, comparing lesbian, gay, bisexual and queer students. The effectiveness of interventions according to demographic factors, such as gender, ethnicity, religion and disability, of students should be explored. Our findings provide guiding principles for schools to develop and implement universal interventions, which could improve inclusivity and acceptance for SGM students and reduce their risk of depression, anxiety, self-harm and suicidality. Our programme theories highlight the importance of the following factors: the overall school climate, including support by school staff and parents, positive representation of SGMs, teacher training and coproduction and codelivery of interventions by SGM students, staff and other experts by lived experience from the wider community. In line with the realist approach, our findings encourage primary research to confirm, refute and refine our theories[28].

## Methods

We used the steps outlined in ref. [26]:

(1) Developed the scope by clarifying the content area
(2) Defined the research questions and ensured there was enough evidence to answer them
(3) Identified how findings and recommendations would be used
(4) Developed search terms and inclusion/exclusion criteria
(5) Identified and screened peer-reviewed papers and data from other sources including websites and grey literature
(6) Extracted and synthesized data
(7) Validated findings with experts by lived experience (see below) to draw inferences and make hypotheses.

We followed RAMESES guidelines for realist reviews (Supplementary Table 1)[76].

### Consultation with experts and reference groups

The review process was guided by a reference group which consisted of a YPAG, a SAG and experts in the field of SGM mental health practice and research. Reference groups and experts help identify relevant sources and fill gaps in programme theories, ensuring the quality of the rapidly produced evidence[26,27]. They further advise on the dissemination and use of findings. The reference group provided information on the relevance and applicability of findings. Full insights from the reference groups are presented in the Supplementary Information.

### Young Person's Advisory Group

The YPAG consisted of eight sexual and/or gender minority young people (aged 14 to 24 years) including those with lived experience of mental health problems. Young people were recruited through the McPin Foundation's Young People's Network, a leading charity placing lived experience at the heart of mental health research. One YPAG member joined our research team and worked on the literature search, data extraction and synthesis. We held three 1.5 h long involvement meetings. Meeting one focused on identifying the content area and defining the research question (steps 1 and 2). The YPAG also advised on search terms and suggested relevant organizations to identify grey literature (steps 4 and 5). Meeting two focused on interpreting preliminary findings and how they could be used in practice (steps 3 and 7). Meeting three focused on validating findings to refine

programme theories (step 7). The YPAG group also advised on the dissemination of findings in the form of a tool kit for schools.

## Stakeholder's Advisory Group
The SAG represented knowledge users and comprised a secondary school governor, a secondary school teacher and two members of the UK government Department for Education. The SAG advised on what currently happens in schools and what would be useful and feasible. We held two meetings to work on steps 1 and 2 and validated findings via email to refine programme theories (steps 3 and 7).

## Experts in SGM mental health research and practice
Our author team consisted of experts by experience (*n* = 3), research (*n* = 2) and clinical practice (*n* = 1). This ensured the consistency of findings with previous literature[26].

## Search strategy
We conducted an exploratory scoping search using Google Scholar to identify key sources and reviews and develop an initial programme theory. For the main search, we searched PubMed, PsycINFO and Web of Science on 14 September 2021. Search terms related to sexual and gender identity (LGBTQ+ OR LGBT* OR LGB* OR queer OR sexual identit* OR sexual orientation OR gender identit* OR lesbian OR gay OR bisexual OR transgender OR nonbinary OR non-binary OR asexual OR pansexual OR sexualit* OR intersex OR omnisexual OR 'questioning sexuality' OR 'questioning gender' OR demisexual OR aromantic) and intervention type (school OR school-based OR educat* AND intervent* OR program* OR polic* OR curricul*). Sexual and gender identity and intervention search terms were combined with the Boolean operator AND (searches for each of the databases in Supplementary Table 11). We restricted the search to titles and abstracts. We consulted experts, the YPAG and relevant organizations to identify grey literature. A call for evidence was disseminated via Twitter to invite schools, organizations and young people to submit evidence.

## Inclusion and exclusion criteria
We included any study design as well as non-peer-reviewed reports posted on websites of relevant LGBTQ+ organizations. There were no restrictions in publication dates but only sources in English were included. We excluded sources that did not provide enough detail to contribute to the development of programme theories.

## Participants
We included sources relating to any sexual and gender identity, including SGM, that is people who are not heterosexual or cisgender, heterosexual and cisgender students who were aged 11–18 years and attending secondary school. We also included sources of secondary school teaching staff. We were primarily interested in universal interventions aimed at all students and teaching staff. We included interventions aimed solely at students or staff. If a source included students under age 11 or above age 18 years, we reviewed its contribution to the programme theory to determine inclusion.

## Main outcome(s)
We included: depression, anxiety, self-harm and suicidality. We also included measures of inclusivity and acceptance: bullying, school climate, school connectedness, stigma, prejudice and discrimination.

## Study selection
We imported records into Rayyan and removed duplicates. Titles and abstracts were split and screened by two researchers (M.S. and T.S.). A 10% random sample was reviewed independently by a third researcher (T.W.). Full texts were split and screened by five researchers. A 10% random sample was reviewed independently by a third researcher (A.P.). Disagreements were resolved by consensus or after discussion with the lead researcher (G. Lewis). Reasons for exclusion were recorded, acknowledging that some records might have multiple reasons for exclusion.

## Data extraction
We used a data extraction schedule to extract: study aim(s) and design, intervention type, sample characteristics and size, context, mechanisms, outcomes and CMO configurations. Contexts comprised information on school setting, intervention type and target group, which may impact outcomes investigated by a source. We identified mechanisms from the quantitative or qualitative analyses that were conducted by the authors of each source, for example through examination of mediator variables or qualitative themes or indirectly in the discussion sections of each source. CMOs were formulated on the basis of the findings presented by the authors of each source or identified by the reviewing team who linked findings with information from the introduction and discussion of each source.

## Data synthesis
We developed an initial programme theory on the basis of key sources from our exploratory scoping search, literature on minority stress theory and discussions with our reference group[76]. We refined the programme theory on the basis of the extracted CMOs from published sources and studies, as well as feedback from the reference group. We grouped the evidence into intervention categories which were informed by our experts, reference groups and literature. The intervention categories were refined throughout the synthesis, to ensure their relevance and applicability. Within each intervention category, CMOs were synthesized on the basis of similar context and mechanism associations, which were linked to outcomes. The reference group identified gaps in these programme theories and highlighted which findings resonated with their lived experience or work. They also provided feedback on the feasibility, implementation and likely effectiveness of interventions. This information was used to expand on contexts and mechanisms.

## Quality assessment
Realist review methodology does not usually recommend a formal quality assessment and focuses instead on the rigour and relevance of sources to the programme theory[26]. During the extraction phase, we assessed each source in terms of whether the evidence contributed to theory development and excluded sources which did not provide sufficient information to extract CMOs. The rigour of sources was assessed on the basis of the credibility and trustworthiness of the methods[76]. To explore rigour, we extracted information on the study design, whether the methodological approach and data collected allowed the study to address the research question within the target population and whether the interpretation of results was sufficiently substantiated by the data (Supplementary Table 3). We also extracted information on the sample size, sampling strategy and adjustment for confounders whilst acknowledging that methodologically weak sources can still provide relevant information for the refinement of programme theories within realist methodologies[29].

## Preregistration
We preregistered our protocol with the prospective register of systematic reviews, PROSPERO: https://www.crd.york.ac.uk/prospero/display_record.php?RecordID=279193. No changes were made to the protocol or reviewing process after registration.

## Inclusion and ethics statement
The review was conducted by academic and lived-experience researchers in the field of SGM mental health. Responsibilities of co-authors were agreed collaboratively ahead of the review. The review draws upon national and international evidence and input from our reference group. No ethical approval was required.

## Reporting summary

Further information on research design is available in the Nature Portfolio Reporting Summary linked to this article.

## Data availability

The data supporting the findings of this study are openly available in the individual sources which constitute the review. Data from the synthesis are available within the review and Supplementary Tables 1–3. Any further details required are available from the corresponding author upon reasonable request. Submission to a public repository is not applicable. We conducted an exploratory scoping search using Google Scholar to identify key sources and reviews and develop an initial programme theory. For the main search, we searched PubMed, PsycINFO and Web of Science (search terms in Supplementary Table 2). We consulted experts, the YPAG and relevant organizations to identify grey literature. A call for evidence was disseminated via Twitter to invite schools, organizations and young people to submit evidence. We used the review software package Rayyan but, as this study was a review, there was no statistical code.

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

## Acknowledgements

This work was funded by a Wellcome Trust Mental Health Active Ingredients commission awarded to G. Lewis as Principal Investigator at University College London. We would like to thank the McPin Foundation, a leading charity placing lived experience at the heart of mental health research, for recruiting our YPAG, convening our involvement meetings and leading our coproduction work. We thank the YPAG for their invaluable insights. We would also like to thank members of our SAG for advising on the development of our programme theory. The funders had no role in study design, data collection and analysis, decision to publish or preparation of the manuscript. G. Lewis is supported by a Sir Henry Dale Fellowship jointly funded by the Wellcome Trust and the Royal Society (grant no. 223248/Z/21/Z). A.P. is supported by Camden and Islington NHS Foundation Trust. A.P. and G. Lewis are also supported by the National Institute for Health Research, University College London Hospital Biomedical Research Centre.

## Author contributions

G. Lewis, T.W., A.P., M.S. and T.S. designed the study. M.L. and T.S. conducted the searches, data extraction and analyses with input from T.W., G. Lewis and G. Levy. G. Lewis, M.L. and D.S. drafted the manuscript and all authors critically reviewed and provided written feedback on drafts. G. Levy was lived-experience consultant and provided critical feedback on each draft.

## Competing interests

The authors declare no competing interests.

## Additional information

**Correspondence and requests for materials** should be addressed to Gemma Lewis.

# Reporting Summary

## Statistics

For all statistical analyses, confirm that the following items are present in the figure legend, table legend, main text, or Methods section.

| n/a | Confirmed | |
|---|---|---|
| ☒ | ☐ | The exact sample size (*n*) for each experimental group/condition, given as a discrete number and unit of measurement |
| ☒ | ☐ | A statement on whether measurements were taken from distinct samples or whether the same sample was measured repeatedly |
| ☒ | ☐ | The statistical test(s) used AND whether they are one- or two-sided <br> *Only common tests should be described solely by name; describe more complex techniques in the Methods section.* |
| ☒ | ☐ | A description of all covariates tested |
| ☒ | ☐ | A description of any assumptions or corrections, such as tests of normality and adjustment for multiple comparisons |
| ☒ | ☐ | A full description of the statistical parameters including central tendency (e.g. means) or other basic estimates (e.g. regression coefficient) AND variation (e.g. standard deviation) or associated estimates of uncertainty (e.g. confidence intervals) |
| ☒ | ☐ | For null hypothesis testing, the test statistic (e.g. *F*, *t*, *r*) with confidence intervals, effect sizes, degrees of freedom and *P* value noted <br> *Give P values as exact values whenever suitable.* |
| ☒ | ☐ | For Bayesian analysis, information on the choice of priors and Markov chain Monte Carlo settings |
| ☒ | ☐ | For hierarchical and complex designs, identification of the appropriate level for tests and full reporting of outcomes |
| ☒ | ☐ | Estimates of effect sizes (e.g. Cohen's *d*, Pearson's *r*), indicating how they were calculated |

*Our web collection on statistics for biologists contains articles on many of the points above.*

## Software and code

Policy information about availability of computer code

| Data collection | As this study was a review, primary data were not collected. The software Rayyan was used for the collection and analysis of data from existing sources (published and unpublished studies and sources). |
|---|---|
| Data analysis | No statistical code was used; Rayyan does not involve coding and no primary data were analysed because this study was a review of existing literature. |

For manuscripts utilizing custom algorithms or software that are central to the research but not yet described in published literature, software must be made available to editors and reviewers. We strongly encourage code deposition in a community repository (e.g. GitHub). See the Nature Portfolio guidelines for submitting code & software for further information.

## Data

Policy information about availability of data

All manuscripts must include a data availability statement. This statement should provide the following information, where applicable:
- Accession codes, unique identifiers, or web links for publicly available datasets
- A description of any restrictions on data availability
- For clinical datasets or third party data, please ensure that the statement adheres to our policy

The data that support the findings of this study are included in Supplementary Tables 1, 2 and 3 and any further details are available from the corresponding author upon reasonable request. All data used in this study are available in the individual studies which constitute the review. Submission to a public repository is not applicable. We conducted an exploratory scoping search using Google Scholar to identify key sources and reviews and develop an initial programme theory. For the main search, we searched PubMed, PsycINFO and Web of Science (search terms in Supplementary Table 2). We consulted experts, the YPAG, and relevant organisations to identify grey literature. A Call for Evidence was disseminated via Twitter to invite schools, organisations, and young people to submit evidence.

# Field-specific reporting

Please select the one below that is the best fit for your research. If you are not sure, read the appropriate sections before making your selection.

☒ Life sciences  ☐ Behavioural & social sciences  ☐ Ecological, evolutionary & environmental sciences

For a reference copy of the document with all sections, see nature.com/documents/nr-reporting-summary-flat.pdf

# Life sciences study design

All studies must disclose on these points even when the disclosure is negative.

| | |
|---|---|
| Sample size | As this was a rapid realist review (with no analysis of aggregate data or meta-analysis), there was no single sample size |
| Data exclusions | As this study was a review there were not data exclusions other than the exclusion criteria for the review itself |
| Replication | Replication was not applicable because this study was a Rapid Realist Review so replication was not required. |
| Randomization | Randomization was not required because this study was a Rapid Realist Review (rather than a trial or any other kind of experimental design). |
| Blinding | Blinding was not required because this study was a Rapid Realist Review (rather than a trial or any other kind of experimental design). |

# Reporting for specific materials, systems and methods

We require information from authors about some types of materials, experimental systems and methods used in many studies. Here, indicate whether each material, system or method listed is relevant to your study. If you are not sure if a list item applies to your research, read the appropriate section before selecting a response.

## Materials & experimental systems

| n/a | Involved in the study |
|---|---|
| ☒ | ☐ Antibodies |
| ☒ | ☐ Eukaryotic cell lines |
| ☒ | ☐ Palaeontology and archaeology |
| ☒ | ☐ Animals and other organisms |
| ☒ | ☐ Human research participants |
| ☒ | ☐ Clinical data |
| ☒ | ☐ Dual use research of concern |

## Methods

| n/a | Involved in the study |
|---|---|
| ☒ | ☐ ChIP-seq |
| ☒ | ☐ Flow cytometry |
| ☒ | ☐ MRI-based neuroimaging |

