## [Peer Review File · Nature Human Behaviour]

Peer Review Information

Journal: Nature Human Behaviour

Manuscript Title: A rapid realist review of universal interventions to promote inclusivity and acceptance of diverse sexual and gender identities in schools

Corresponding author name(s): Gemma Lewis

Reviewer Comments & Decisions:

Decision Letter, initial version:
--

25th August 2022

Dear Dr Lewis,

Thank you once again for your manuscript, entitled "Promoting inclusivity and acceptance of diverse sexual and gender identities in schools, to improve young people's mental health: Rapid Realist Review," and for your patience during the peer review process. Please accept our sincere apologies for the delay in reaching a decision.

Your manuscript has now been evaluated by 2 reviewers, whose comments are included at the end of this letter. Although the reviewers find your work to be of interest, they also raise some important concerns. We are very interested in the possibility of publishing your study in Nature Human Behaviour, but would like to consider your response to these concerns in the form of a revised manuscript before we make a decision on publication.

To guide the scope of the revisions, the editors discuss the referee reports in detail within the team, including with the chief editor, with a view to (1) identifying key priorities that should be addressed in revision and (2) overruling referee requests that are deemed beyond the scope of the current study. We hope that you will find the prioritised set of referee points to be useful when revising your study. Please do not hesitate to get in touch if you would like to discuss these issues further.

1. Reviewer 1 requests that you follow RAMESES guidelines and asks that you include rigour of the research studies in your analysis and synthesis. In your revised manuscript, please follow RAMESES guidelines as well as assess and include rigour in your analysis.

2. The reviewers ask that you motivate your choice to conduct a rapid realist review and explicitly discuss why this approach is suitable to address your research question. In addition, the reviewers would like you elaborate on the methodology and clarify methodological choices. Please carefully

evaluate these requests and address them in full.

In sum, we invite you to revise your manuscript taking into account all reviewer and editor comments. We are committed to providing a fair and constructive peer-review process. Do not hesitate to contact us if there are specific requests from the reviewers that you believe are technically impossible or unlikely to yield a meaningful outcome.

We hope to receive your revised manuscript within two months. I would be grateful if you could contact us as soon as possible if you foresee difficulties with meeting this target resubmission date.

- Include a "Response to the editors and reviewers" document detailing, point-by-point, how you addressed each editor and referee comment. If no action was taken to address a point, you must provide a compelling argument. When formatting this document, please respond to each reviewer comment individually, including the full text of the reviewer comment verbatim followed by your response to the individual point. This response will be used by the editors to evaluate your revision and sent back to the reviewers along with the revised manuscript.
- Highlight all changes made to your manuscript or provide us with a version that tracks changes.

[REDACTED]

We look forward to seeing the revised manuscript and thank you for the opportunity to review your work. Please do not hesitate to contact me if you have any questions or would like to discuss these revisions further.

Sincerely,

Samantha Antusch

Samantha Antusch, PhD
Editor
Nature Human Behaviour

Reviewer expertise:

Reviewer #1: diverse sexual and gender youth ; rapid realist reviews

Reviewer #2: rapid realist reviews

REVIEWER COMMENTS:

Reviewer #1:
Remarks to the Author:

Dear authors,

Thank you for the opportunity to review the article, "Promoting inclusivity and acceptance of sexual minority and trans young people in schools to improve mental health: a Rapid Realist Review of universal interventions", which offers a timely and useful contribution to the evidence base on preventing mental health inequities experienced by these communities linked to prevailing social norms. In particular, by offering insights into the contexts in which potential public health interventions may work, for whom and how – which helps inform planning, research and evaluation of such initiatives. Please find below some minor comments/suggestions to help strengthen/clarify in places what is otherwise a well written report making excellent use of engagement.

Abstract line 31 – amend to could 'help' or 'contribute to' preventing mental health problems

Abstract lines 46 /54 – amend to sexual minority and 'gender diverse' young people, to acknowledge that not all gender diverse people identify as trans (e.g. some non-binary people do not identify as trans)?

Intro – Just a suggestion but it helps to change narrative away from LGBTQ+ being a problem/inherently linked to poor mental health by contextualising elevated rates of MH with the risk factors and theory underpinning this first before identifying the elevated rates of mental ill health. The authors did not follow RAMESES guidance for realist syntheses, which may be useful to help give more detail in places, e.g. around the conceptualisation of mechanisms and context:
https://www.ramesesproject.org/media/RS_qual_standards_researchers.pdf

Methods

Excellent incorporation of stakeholder engagement and advisory groups which is great to see and essential in this avenue of research.

For readers not familiar with realist evaluation/synthesis more information would be helpful to inform how the initial programme theories were devised from the individual studies.

It would also be helpful to be explicit about how the rapid realist review differs from 'standard' realist synthesis to help the reader gauge the appropriateness of methodology, e.g. usually initial programme theories are developed prior to the search phase and then refined iteratively throughout the review. Then the results section would refer to the initial hypotheses and the extent to which they are supported or not.

Quality assessment – Realist syntheses usually appraise potential sources of information in relation to both relevance (which the authors have referenced) and rigour 'whether the method used to generate that particular piece of data is credible and trustworthy' and this information is taken into account during analysis and synthesis. (see Wong G, Greenhalgh T, Westhorp G, Pawson R..Development of methodological guidance, publication standards and training materials for realist and meta-narrative reviews: the RAMESES (Realist And Meta-narrative Evidence Syntheses - Evolving Standards) project. Health Serv Deliv Res 2014;2(30))

Data extraction – more information is needed here if the study were to be replicable, it says CMOs were extracted from studies but studies would not have presented CMOs so this is insufficient to explain what information was extracted, how it contributed to theory development and refinement etc. Line 254 – the use of the term 'normalise LGBTQ+' identities is contested as it suggests that they are currently not 'normal'.

The potential for harm/unintended consequences sections of the results sections are helpful.

Discussion

benefit from implications and conclusions section

Reviewer #2:

Remarks to the Author:

This paper focuses on a rapid realist review (RR) of inclusivity and acceptance of sexual minority and trans young people in schools to improve mental health. I wish to congratulate the authors on undertaking this timely work.

Some areas I would like the authors to refine/provide more details:

Why the use of RRR-Needs to be further expanded-cite evidence it being used, and why it's useful for your question.

Methods

The methods need to be enhanced.

I note the use of Saul, but I would like some more descriptive background as to why this approach is useful.

Saul also advocates for reference panels this needs to be clarified in the methods linked to the YPAG & SAG -also, were these groups involved in refining the question?

Consensus on the papers-what was the process of developing the CMO's?

The summary of the papers is good-would like to see a discussion on the use of RRR approach for this context.

Author Rebuttal to Initial comments

Response to reviewers:

Reviewer #1:

Remarks to the Author:

Dear authors,

Thank you for the opportunity to review the article, “Promoting inclusivity and acceptance of sexual minority and trans young people in schools to improve mental health: a Rapid Realist Review of universal interventions”, which offers a timely and useful contribution to the evidence base on preventing mental health inequities experienced by these communities linked to prevailing social norms. In particular, by offering insights into the contexts in which potential public health interventions may work, for whom and how – which helps inform planning, research and evaluation of such initiatives. Please find below some minor comments/suggestions to help strengthen/clarify in places what is otherwise a well written report making excellent use of engagement.

1. Abstract line 31 – amend to could ‘help’ or ‘contribute to’ preventing mental health problems

We have made this change to the abstract (page 2).

2. Abstract lines 46 /54 – amend to sexual minority and ‘gender diverse’ young people, to acknowledge that not all gender diverse people identify as trans (e.g. some non-binary people do not identify as trans)?

In line with recommendations from the reviewer and our lived experience researchers, we have changed the term ‘trans’ to ‘gender minorities’ in the revised manuscript. We agree that, whilst the term ‘trans’ is often used for any gender that is not cis (Matsuno et al. 2022; Mayer et al. 2008), not all gender diverse or non-binary people identify as trans. However, our lived

experience researcher (a transgender woman) pointed out that not all transgender people identify as gender diverse. We have therefore chosen ‘gender minority’ to refer to anyone who does not identify as cis. We now use ‘sexual and gender minorities (SGM)’ to describe anyone who does not belong to the sexual or gender majority (Flatt et al. 2022).

3. Intro – Just a suggestion but it helps to change narrative away from LGBTQ+ being a problem/inherently linked to poor mental health by contextualising elevated rates of MH with the risk factors and theory underpinning this first before identifying the elevated rates of mental ill health.

In the revised manuscript, we now describe minority stress theory and associated risk factors first (page 3). We then describe evidence on the elevated rates of mental health problems in sexual and gender minorities compared with heterosexual and cisgender people (page 3).

4. The authors did not follow RAMESES guidance for realist syntheses, which may be useful to help give more detail in places, e.g. around the conceptualisation of mechanisms and context: https://www.ramesesproject.org/media/RS_qual_standards_researchers.pdf

We have followed RAMESES guidelines and completed the RAMESES checklist in the revised manuscript (Supplementary Table 1). We have described our approach in more detail in the methods, including the conceptualisation of mechanisms and contexts under data extraction (page 21).

Methods

Excellent incorporation of stakeholder engagement and advisory groups which is great to see and essential in this avenue of research.

Thank you very much.

For readers not familiar with realist evaluation/synthesis more information would be helpful to inform how the initial programme theories were devised from the individual studies.

We have added information on the data extraction and synthesis (page 21-22) and explained how initial programme theories were devised from individual studies (page 21-22).

It would also be helpful to be explicit about how the rapid realist review differs from ‘standard’ realist synthesis to help the reader gauge the appropriateness of methodology, e.g. usually initial programme theories are developed prior to the search phase and then refined iteratively

throughout the review. Then the results section would refer to the initial hypotheses and the extent to which they are supported or not.

We have added information on how rapid realist reviews differ from traditional realist reviews to the introduction (page 4-5). We now explain that rapid realist reviews not only involve reference groups but also experts in the topic (page 4-5). We have elaborated on the role of experts and reference groups in our methods as rapid realist reviews rely more heavily on their involvement compared to traditional realist reviews (page 18-19). We have now explained that we developed an initial programme theory (previously referred to as hypothesis) based on previous literature and discussions (page 22).

Quality assessment – Realist syntheses usually appraise potential sources of information in relation to both relevance (which the authors have referenced) and rigour ‘whether the method used to generate that particular piece of data is credible and trustworthy’ and this information is taken into account during analysis and synthesis. (see Wong G, Greenhalgh T, Westhorp G, Pawson R..Development of methodological guidance, publication standards and training materials for realist and meta-narrative reviews: the RAMESES (Realist And Meta-narrative Evidence Syntheses - Evolving Standards) project. Health Serv Deliv Res 2014;2(30))

We have appraised the rigour of research studies in our data analysis and synthesis in the revised manuscript. We have added a table containing a rigour assessment of the included studies (supplementary Table 3). We have added a discussion of the rigour assessment to the methods (page 22-23) and a summary of the rigour of the studies to the results (page 5) and discussion (page 15).

The rigour of sources was assessed based on the credibility and trustworthiness of the methods, in line with guidelines for Rapid Realist reviews (Wong et al., 2013). To explore rigour, we extracted information on the study design, whether the methodological approach and data collected allowed the study to address the research question within the target population, and whether the interpretation of results was sufficiently substantiated by the data (Supplementary Table 3). We also extracted information on the sample size, sampling strategy, and adjustment for confounders whilst acknowledging that methodologically weak sources can still provide relevant information for the refinement of programme theories within realist methodologies (Pawson & Tilley, 1997).

Data extraction – more information is needed here if the study were to be replicable, it says CMOs were extracted from studies but studies would not have presented CMOs so this is

insufficient to explain what information was extracted, how it contributed to theory development and refinement etc.

We have elaborated on our data extraction and synthesis to facilitate the replication of our review (pages 21-22 and see response above).

Line 254 – the use of the term ‘normalise LGBTQ+’ identities is contested as it suggests that they are currently not ‘normal’.

We have changed the term ‘normalise to ‘validate throughout the manuscript.

The potential for harm/unintended consequences sections of the results sections are helpful.

Thank you very much.

Discussion

benefit from implications and conclusions section

We have expanded upon our implications and conclusions section and re-structured the discussion so that these sections are more clearly located (page 15-17).

Reviewer #2:

Remarks to the Author:

This paper focuses on a rapid realist review (RR) of inclusivity and acceptance of sexual minority and trans young people in schools to improve mental health. I wish to congratulate the authors on undertaking this timely work.

Some areas I would like the authors to refine/provide more details:

Why the use of RRR-Needs to be further expanded-cite evidence it being used, and why it's useful for your question.

We have expanded upon our rationale for conducting a RRR to answer our research questions and cited evidence of the approach previously being used (page 4-5).

Methods

The methods need to be enhanced.

I note the use of Saul, but I would like some more descriptive background as to why this approach is useful.

We have added information on the rationale behind using a RRR and the usefulness of this approach (page 5). The methods section has also been enhanced by our inclusion of the RAMESES guidelines and an assessment of the rigour of research studies in our analysis and synthesis (see response above).

Saul also advocates for reference panels this needs to be clarified in the methods linked to the YPAG & SAG -also, were these groups involved in refining the question?

We have clarified the involvement of the reference group and the experts within our author group (introduction; page 4-5 and methods; page 18-19). We now explain that these groups were involved in refining the research question (page 18-19).

We have also clarified that our author team was carefully selected based on each author's expertise, i.e. experts by experience of being sexual and gender minorities, as well as experts in research on mental health among sexual and gender minorities, and clinical practice (page 19).

Consensus on the papers-what was the process of developing the CMO's?

We have clarified the process of extracting and synthesising CMOs in our methods section (page 21-22).

The summary of the papers is good-would like to see a discussion on the use of RRR approach for this context.

We now discuss the use of the RRR approach in the summary of studies section by addressing the relevance and rigour of studies and discussing the study designs and inclusion of other, non-peer reviewed sources following RRR principles. We have added a table containing a rigour assessment of the included studies (Supplementary Table 3), and discussed this in the methods (page 22). We have summarised the rigour of the studies in the results (page 5) and discussion (page 15).

References

- Wong G, Greenhalgh T, Westhorp G, Buckingham J, Pawson R. RAMESES publication standards: realist syntheses. *J Adv Nurs* 2013; 69: 1005–22.
- Pawson R, Tilley Nick. *Realistic evaluation*. Sage, 1997.
- Matsuno, E., Goodman, J. A., Israel, T., Choi, A. Y., Lin, Y. J., & Kary, K. G. (2022). L or g or b or t: Matching sexual and gender minorities with subpopulation-specific interventions. *Journal of Homosexuality*, 69(3), 385-407.
- Mayer, K. H., Bradford, J. B., Makadon, H. J., Stall, R., Goldhammer, H., & Landers, S. (2008). Sexual and gender minority health: what we know and what needs to be done. *American journal of public health*, 98(6), 989-995.
- Flatt, J. D., Cicero, E. C., Kittle, K. R., & Brennan-Ing, M. (2022). Recommendations for advancing research with sexual and gender minority older adults. *The Journals of Gerontology: Series B*, 77(1), 1-9.

Decision Letter, first revision:

29th November 2022

Dear Dr. Lewis,

Thank you for submitting your revised manuscript "Universal interventions to promote inclusivity and acceptance of diverse sexual and gender identities in schools, a Rapid Realist Review" (NATHUMBEHAV-22041073A). It has now been seen by the original referees and their comments are below. As you can see, the reviewers find that the paper has improved in revision. We will therefore be happy in principle to publish it in *Nature Human Behaviour*, pending minor revisions to satisfy the referees' final requests and to comply with our editorial and formatting guidelines.

We are now performing detailed checks on your paper and will send you a checklist detailing our editorial and formatting requirements within a week. Please do not upload the final materials and make any revisions until you receive this additional information from us.

Sincerely,

Samantha Antusch

Samantha Antusch, PhD

Senior Editor
Nature Human Behaviour

Reviewer #2 (Remarks to the Author):

Thank you for the opportunity to re-review this rapid realist synthesis of schools based interventions to support LGBTQ+ student mental health which is a timely and important contribution to the literature on this topic. The authors have responded to the reviewer suggestions and as a result the article is strengthened. However, there are a few outstanding issues affecting clarity and readability:

- there is no methods section in text and the information at the end is not referenced in the text so it is unclear until the that there is more detail provided in appendices, e.g. which databases were searched, what search terms were used, with what parameters, how was rigour assessed, which groups have been included and how?? I can see supplementary tables are included but there needs to be more information in text so that readers can understand how the results came about as they read through.

- Spell out all abbreviations at first use in text e.g. YPAG, SAG, RCT etc

- It's is unclear from the methods/results in the text when reading through whether the refined programme theories for each intervention are developed from information in the studies published (e.g. through examination of mediator variables, qualitative themes etc) or from feedback from YPAG/SAG (which have not been described anywhere), or a mixture of both? How was this process arrived at?

Minor amendments to signpost reader more clearly to the methodology would address this issue. Thanks again for the invitation to review this insightful review.

Reviewer #3 (Remarks to the Author):

The authors have satisfied my queries and the paper has been refined.

Final Decision Letter:

Dear Dr Lewis,

We are pleased to inform you that your Article "A rapid realist review of universal interventions to promote inclusivity and acceptance of diverse sexual and gender identities in schools", has now been accepted for publication in Nature Human Behaviour.

Please note that *Nature Human Behaviour* is a Transformative Journal (TJ). Authors whose manuscript

was submitted on or after January 1st, 2021, may publish their research with us through the traditional subscription access route or make their paper immediately open access through payment of an article-processing charge (APC). Authors will not be required to make a final decision about access to their article until it has been accepted. IMPORTANT NOTE: Articles submitted before January 1st, 2021, are not eligible for Open Access publication. Find out more about Transformative Journals

Once your manuscript is typeset and you have completed the appropriate grant of rights, you will receive a link to your electronic proof via email with a request to make any corrections within 48 hours. If, when you receive your proof, you cannot meet this deadline, please inform us at risproduction@springernature.com immediately. Once your paper has been scheduled for online publication, the Nature press office will be in touch to confirm the details.

We welcome the submission of potential cover material (including a short caption of around 40 words) related to your manuscript; suggestions should be sent to Nature Human Behaviour as electronic files (the image should be 300 dpi at 210 x 297 mm in either TIFF or JPEG format). Please note that such pictures should be selected more for their aesthetic appeal than for their scientific content, and that colour images work better than black and white or grayscale images. Please do not try to design a cover with the Nature Human Behaviour logo etc., and please do not submit composites of images related to

your work. I am sure you will understand that we cannot make any promise as to whether any of your suggestions might be selected for the cover of the journal.

With best regards,

Samantha Antusch

Samantha Antusch, PhD
Senior Editor
Nature Human Behaviour